Effects of tire leachate on the invasive mosquito Aedes albopictus and the native congener Aedes triseriatus

Villena Oswaldo C. 1
Terry Ivana 1
Iwata Kayoko 2
Landa Edward R. 1
LaDeau Shannon L. 3
Leisnham Paul T. leisnham@umd.edu 1
1 Department of Environmental Science and Technology, University of Maryland , College Park , MD , United States of America
2 Graduate School of Agriculture, Kyoto University , Kyoto , Japan
3 Cary Institute of Ecosystem Studies , Millbrook , NY , United States of America
Negri Ilaria
Electronic publication date: 2017 Sep 5
Publication date: 2017
Volume: 5
Electronic Location ID: e3756
Received 2017 Jun 29; Accepted 2017 Aug 12
Copyright: ©2017 Villena et al.
Copyright year: 2017
Copyright holder: Villena et al.
License: This is an open access article distributed under the terms of the Creative Commons Attribution License, which permits unrestricted use, distribution, reproduction and adaptation in any medium and for any purpose provided that it is properly attributed. For attribution, the original author(s), title, publication source (PeerJ) and either DOI or URL of the article must be cited.
License URL: https://creativecommons.org/licenses/by/4.0/

Keywords: Competition, Invasion biology, Pest control, Tire leachate, Urbanization, West nile virus, Zinc, Toxins, Asian tiger mosquito

Funding: Maryland Agriculture Experimental Station MD-ENST-2956 NSF-Couple Natural Human Systems Program DEB-1211797 This research was funded by a Maryland Agriculture Experimental Station Grant MD-ENST-2956 to Paul T. Leisnham and an NSF-Couple Natural Human Systems Program (DEB-1211797) to Shannon LaDeau and Paul Leisnham. The funders had no role in study design, data collection and analysis, decision to publish, or preparation of the manuscript.

==============================
Discarded vehicle tire casings are an important artificial habitat for the developmental stages of numerous vector mosquitoes. Discarded vehicle tires degrade under ultraviolet light and leach numerous soluble metals (e.g., barium, cadmium, zinc) and organic substances (e.g., benzothiazole and its derivatives [BZTs], polyaromatic hydrocarbons [PAHs]) that could affect mosquito larvae that inhabit the tire casing. This study examined the relationship between soluble zinc, a common marker of tire leachate, on mosquito densities in tire habitats in the field, and tested the effects of tire leachate on the survival and development of newly hatched Aedes albopictus and Aedes triseriatus larvae in a controlled laboratory dose-response experiment. In the field, zinc concentrations were as high as 7.26 mg/L in a single tire and averaged as high as 2.39 (SE ± 1.17) mg/L among tires at a single site. Aedes albopictus (37/42 tires, 81.1%) and A. triseriatus (23/42, 54.8%) were the most widespread mosquito species, co-occurred in over half (22/42, 52.4%) of all tires, and A. triseriatus was only collected without A. albopictus in one tire. Aedes triseriatus was more strongly negatively associated with zinc concentration than A. albopictus, and another common mosquito, C. pipiens, which was found in 17 tires. In the laboratory experiment, A. albopictus per capita rate of population change (λ′) was over 1.0, indicating positive population growth, from 0–8.9 mg/L zinc concentration (0–10,000 mg/L tire leachate), but steeply declined to zero from 44.50–89.00 mg/L zinc (50,000–100,000 mg/L tire leachate). In contrast, A. triseriatus λ′ declined at the lower concentration of 0.05 mg/L zinc (100 mg/L tire leachate), and was zero at 0.45, 8.90, 44.50, and 89.00 mg/L zinc (500, 10,000, 50,000 and 100,000 mg/L tire leachate). These results indicate that tire leachate can have severe negative effects on populations of container-utilizing mosquitoes at concentrations commonly found in the field. Superior tolerance to tire leachate of A. albopictus compared to A. triseriatus, and possibly other native mosquito species, may have facilitated the replacement of these native species as A. albopictus has invaded North America and other regions around the world.

Introduction

The distributions and abundances of adult mosquitoes are strongly affected by processes occurring at the developmental (eggs, larval) life stages (Washburn, 1995; Juliano, 2009). Larval densities in developmental habitats, including water-holding containers, are regulated by many biotic and physical factors, including resource competition, predation, temperature, and environmental toxins (Clements, 1992; Clements, 1999). Not surprisingly, almost all the literature examining the responses of mosquitoes to environmental toxins is focused on commercial or easily sourced chemicals that may be used as mosquitocides to control vector and nuisance species (Floore, 2006; Tisgratog et al., 2016). However, many soluble chemicals in mosquito developmental habitats may occur at concentrations that are toxic to larvae. For example, in container habitats that receive high amounts of allochthonous leaf litter, foliar tannins can suppress mosquito survival and development, and are likely to alter local community composition (Sota, 1993; Mercer & Anderson, 1994; Smith et al., 2013).

Discarded vehicles are an important artificial habitat for the developmental stages of numerous vector mosquitoes (Yee, 2008). Rubber Manufacturers Association (2014) estimates that there are approximately 75 million stockpiled tires in the United States. Tires are important because they are often in close proximity to human habitation and support higher abundances of larvae than other, usually smaller, developmental habitats (e.g., discarded food containers, plastic trash, bird baths) (Carrieri et al., 2003; Leisnham et al., 2006; Dowling et al., 2013). Furthermore, tires are a common means of accidentally transporting native and invasive species around the United States and the world (Lounibos et al., 2002). Tires are a conglomerate of chemical compounds that degrade under ultraviolet (UV) light and leach numerous soluble metals (e.g., barium, cadmium, zinc) and organic (e.g., benzothiazole and its derivatives [BTs], polyaromatic hydrocarbons [PAHs]) compounds (Wik & Dave, 2009; Wik et al., 2009). A growing body of literature has demonstrated acute toxicity of tire particles, leachate, or specific compounds on living organisms, and in particular aquatic taxa (reviewed by Wik & Dave, 2009). Fewer studies have researched sub-lethal impacts of tire materials (e.g., Draper & Robinson, 2001; Suwanchaichinda & Brattsten, 2002; Gualtieri et al., 2005), and to our knowledge, no studies have tested the effects of tire leachate on the survival and overall population performances of mosquito species.

Aedes albopictus, the Asian tiger mosquito, was introduced to the continental United States in the mid-1980s through shipments of eggs and larvae in used tires (Tatem, Hay & Rogers, 2006). It has since spread throughout the eastern part of the country where its range overlaps with the native treehole mosquito, Aedes triseriatus. Both A. albopictus and A. triseratus commonly inhabit tire habitats (Yee, 2008). Laboratory and field experiments have shown A. albopictus to be competitively superior to A. triseriatus under most conditions (e.g., Livdahl & Willey, 1991; Teng & Apperson, 2000; Aliabadi & Juliano , 2002). Ecological theory and empirical work suggests that with one limiting factor in a constant environment, interspecific competition should result in competitive exclusion (Tilman, 1982; Chase & Leibold, 2003). Yet A. triseriatus has persisted in many areas despite the spread of A. albopictus (Lounibos et al., 2001). One hypothesis for the coexistence of A. triseriatus with A. albopictus may be superior tolerance of the native to environmental toxins, including those in important developmental habitats such as tires. Inferior competitors can escape local extinction by a number of mechanisms, including superior tolerance to environmental conditions, and trade-offs between competitive ability and tolerances to environmental conditions have been well documented in the ecological literature (e.g., Chesson, 1986; Chesson, 2000; Dunson & Travis, 1991; Chesson & Huntly, 1997).

In this study, we will examine the relationship between tire leachate and densities of mosquitoes in discarded tires in the field, and test the effects of tire leachate on the survival and development of A. albopictus and A. triseriatus larvae in a controlled laboratory dose–response study. Zinc has been implicated as an important tire toxicant to aquatic organisms and is one of the most common compounds in tire material (Wik et al., 2009), but a range of potentially organic compounds are present in tire leachate that may also induce acute toxicity or sub-lethal impacts (Wik et al., 2009). Therefore, in this study we will use zinc concentration as a marker of tire leachate consistent with past studies (Wik et al., 2009). In our laboratory study, we will examine the effects of tire leachate based on per capita rate of population change (r, Goldberg & Fleetwood, 1987). In mosquito experiments, population performance can be estimated by calculating an estimate of the finite rate of population change (λ′), which is a composite index based on individual fitness parameters: survivorship, female development time, and female wing length (as a fecundity surrogate).

Materials and Methods

Field study

Mosquitoes were sampled from 42 tires among five sites (6–14 tires per site) in College Park (lat.: 38.993, lon.: −76.955) and Baltimore (lat.: 39.287, lon.: −76.631), Maryland, USA. All sampling was conducted from July to August 2011, which is the period of peak mosquito activity in the area (Dowling et al., 2013). Weather stations within 5 miles of the study sites recorded mean maximum daily temperature of 88 °C and total precipitation of 214.6 mm during the sampling period (National Oceanic and Atmospheric Administration, 2017), which is conducive for mosquito activity. All sites were within predominantly residential and low-development commercial areas. Three sites were where tires had been illegally dumped and two sites were a part of stacked tire piles at auto-repair shops (Table 1). The entire contents of each tire (i.e., water, detritus, and all biota) at each site were collected. All immature mosquito individuals (larvae and pupae) were identified to species level and counted. Detritus was dried at 35 °C for >48 h and weighed as a broad measure of resource quantity. A 75 mL water subsample was acidified at pH 2.0 and refrigerated for later analysis of total nitrogen (TN) and total phosphorus (TP) using Hach test kits (TNT826 and TNT844, respectively) and a Hach 3800 spectrophotometer (Hach Company, Colorado, USA). A second water subsample was filtered immediately after field collection through a syringe-mounted, 0.45 µm pore-size nylon filter. The filtrate was acidified with double distilled nitric acid prior to analysis for zinc using a PerkinElmer Optima 4300 DV Inductively Coupled Plasma-Optical Emissions Spectrophotometer (ICP-OES) (PerkinElmer Inc.; Waltham, Massachusetts, USA)

Table 1 Mean ± SE (range in parentheses) A. albopictus density, A. triseriatus density, C. pipiens density, soluble zinc concentration, detritus amount, total nitrogen concentration, and total phosphorus concentration among tires in six sample sites.

Site	Site type	Number of tires	A. albopictus per tire	A. triseriatus per tire	Cx. pipiens per tire	Soluble zinc (mg/L)	Detritus (g)	Total nitrogen (mg/L)	Total phosphorus (mg/L)	
1	Dump	6	161.7 ± 94.6
(3–489)	20.3 ± 10.4
(1–67)	192.8 ± 142.4
(0–887)	1.37 ± 0.81
(0.05–4.85)	10.56 ± 3.44
(3.35–25.17)	4.08 ± 4.05
(0.00–12.18)	1.76 ± 0.81
(0.38–3.89)	
2	Auto repair shop	10	37.5 ± 12.3
(1–135)	10.0 ± 4.5
(0–43)	23.5 ± 22.9
(0–230)	0.09 ± 0.04
(0.05–0.45)	6.26 ± 1.38
(1.62–14.06)	7.76 ± 1.72
(3.12–18.10)	6.29 ± 3.59
(0.45–34.60)	
3	Dump	14	27.0 ± 8.1
(0–91)	2.7 ± 1.8
(0–24)	8. 8 ± 8.7
(0–122)	1.35 ± 0.26
(0.19–3.14)	2.43 ± 0.42
(0.00–5.34)	4.88 ± 0.48
(1.35–9.42)	0.38 ± 0.04
(0.08–0.65)	
4	Auto repair shop	6	191.0 ± 56.3
(28–367)	23.5 ± 12.5
(0–81)	85.3 ± 61.3
(1–385)	0.43 ± 0.13
(0.05–0.97)	5.4 ± 1.9
(1.03–14.21)	2.82 ± 1.15
(0.38–7.16)	0.85 ± 0.12
(0.65–1.32)	
5	Dump	6	12.5 ± 11.3
(0–69)	2.3 ± 2.0
(0–12)	0.0 ± 0.0
(0)	2.39 ± 1.17
(0.09–7.26)	0.68 ± 0.4
(0.00–2.39)	4.32 ± 1.31
(1.97–9.73)	0.26 ± 0.07
(0.12–0.56)	

Tire leachate preparation

Tire leachate was prepared by extracting ground tire material, minus 30 mesh (less than 0.59 mm) 99.51% dry content, with deionized water (DI) in either Teflon or Teflon coated polypropylene bottles at a solid to liquid ratio of 1:10 (100 g of ground tire material to a liter of DI water). A blank, containing only DI water was also prepared. The samples were shaken for a one-week period on an orbital table shaker at approximately 100 rpm at room temperature (19–22 °C). At the conclusion of the one-week period, the samples were filtered in a class 1,000 laminar flow clean bench through a sterile 0.2 µm filter, into a sterile 1,000 ml polycarbonate container. The filter membrane was surfactant-free Cellulose Acetate (CA); this membrane has no wetting agents that might affect sensitive cell culture lines and is cleaner than ordinary CA membranes for tissue culture use. The sample in the sterile container was bagged and refrigerated prior to usage. For conformity and comparison with earlier studies in the literature, this stock solution was designated as 100% tire leachate at a concentration of 100,000 mg/L. Analysis on this 100% solution using an ICP-OES indicated that it had a zinc concentration of 89 mg/L.

Laboratory experiment

A dose-response laboratory experiment was conducted on A. albopictus and A. triseriatus, which were the two most widespread mosquito species from our field study (see Results), and for which we also had readily available individuals. Aedes albopictus and A. triseriatus eggs sourced from F1−2 colonies at University of Maryland were synchronously hatched in nutrient broth solutions in plastic trays. Within 24 h, 25 newly hatched larvae of each species were rinsed and added to 400 ml cups containing one of six concentrations of tire leachate using dilutions of the 100% stock solution and DI water. The six dilutions and their zinc concentrations (mg/L) were: 100.0% (89.000 mg/L), 50.0% (44.500 mg/L), 10.0% (8.900 mg/L), 1% (0.890 mg/L), 0.5% (0.445 mg/L), and 0.1% (0.089 mg/L). Each concentration × species combination had five replicates to yield 70 total experimental cups (7 concentrations × 2 species × 5 replicates). Cups were housed in an incubator at 25 °C and 16:8 h light-dark cycle. Each cup received 0.1 mg of bovine liver powder:lactalbumin (diluted 1:10). Pupae were removed from containers daily and placed into individual vials until adult emergence. Adults were sexed, identified, dried (>24 h, 50 °C) and weighed, and females’ wings were measured. For each cup, proportion survivorship, median female development time, and median female mass were calculated. These fitness parameters were used to estimate the finite rate of population growth for each species λ′, (Juliano, 1998): λ′= expln1∕N0 ∑xAxfwxD+∑xxAxfwx∑xAxfwx

where N0 is the initial number of females (assumed to be 50% per microcosm), x is the mean time to eclosion (measured in days), Ax is the mean number of females eclosing on day x, wx is the mean body size on day x, and f(wx) is a function describing size dependent fecundity for each species, estimated from the mean wing length on day x, wx, of female mosquitoes (Livdahl & Sugihara, 1984; Juliano, 1998). The function for A. albopictus was f(wx) =  − 121.240 + 78.02wx, where wx is wing length (millimeters) (Lounibos et al., 2002). The function for A. triseriatus was f(wx) = (1∕2)exp[4.5801 + 0.8926(lnwx)] − 1 (Nannini & Juliano, 1998). D is the mean days it takes for an adult mosquito to mate, bloodfeed, and oviposit, and is estimated at 14 days for A. albopictus (Lounibos et al., 2002) and 12 days for A. triseriatus (Nannini & Juliano, 1998).

Statistical analyses

Relationships between zinc concentrations and field abundances of each of the three collected mosquito species: A. albopictus, A. triseriatus, and C. pipiens (see Results) were analyzed using negative binomial regressions (PROC GENMOD, SAS Institute). Because a species’ abundance in a tire may be predicted by various physiochemical and biological container characteristics as well as surrounding landscape variables, we also included site, detritus amount, TN, TP, and the abundances of the other two mosquito species in an initial multi-factor model for each species. Final multi-factor models were selected using backward selection. If there was no significant loss of fit, as evaluated by AIC, we continued to eliminate the next least significant variable until all non-significant variables were removed or until there was significant loss of model fit (e.g., −2 ΔAIC). Correlations among predictor variables were tested using Pearson or Spearman rank correlations (SAS PROC CORR) and multicolinearity was checked by means of Variance Inflation Characteristics (VIF) (SAS PROC REG), with a VIF above 5 for a variable indicating a problem (Kutner, Nachtsheim & Neter, 2004); however, none were evident. We tested for effects of tire leachate concentration and mosquito species from the laboratory experiment using randomization ANOVAs (Randomization wrapper for SAS PROCs; Cassell, 2011) because data (survival, mean development time, mean body size, λ′) failed to meet parametric assumptions despite transformations, and generalized models using variable distribution models (e.g., Poisson, negative binomial) routinely failed to converge. Tire leachate concentration was treated as a categorical variable because preliminary data plots clearly showed that its relationship with each response variable was not linear. For all analyses, randomization ANOVA yielded conclusions identical to those of parametric ANOVA, probably because trends among treatment groups were clear (see Results). For brevity and accuracy, we report only the results from randomization ANOVA. Experimentwise α = 0.05 for all statistical analyses.

Results

Field survey

Aedes albopictus, A. triseriatus and C. pipiens were collected from 88% (37/42), 55% (23/42), and 41% (17/42) of tires, and constituted 55% (2941/5383), 8% (415/5383), and 38% (2027/5383) of total immatures, respectively. Aedes albopictus and A. triseriatus co-occurred in individual tires the most frequently among all pairwise species combinations, being collected together in 22 tires. Aedes triseriatus was only collected without A. albopictus in one tire. Aedes albopictus and A. triseriatus co-occurred with C. pipiens in 17 and 15 tires, respectively. There was considerable variation among individual tires and sites in mosquito abundances and physiochemical characteristics of tires (Table 1). Zinc concentrations were as high as 7.26 mg/L, recorded in a single tire at Site 5, and as low as 0.05 mg/L, recorded at Sites 1, 2, and 4 (Table 1). The three dump sites had higher average zinc concentrations compared to the two sites at auto-repair shops (Table 1). In final multi-factor models, A. triseriatus (Estimate: −0.8019; χ12=5.37, P = 0.0204) was most strongly negatively associated with zinc concentration, followed by A. albopictus (Estimate: −0.6340; χ12=6.51, P = 0.0107), and C. pipiens (Estimate: −0.2190; χ12=5.02, P = 0.0251). Additionally, A. albopictus was associated positively with detritus amount (Estimate: 0.6340; χ12=5.37, P = 0.0204); A. triseriatus was positively associated with A. albopictus (Estimate: 0.0112; χ12=6.34, P = 0.0118) and C. pipiens (Estimate: 0.0092; χ12=5.74, P = 0.0166); and C. pipiens varied with site (χ12=13.79, P = 0.0080) and was positively associated to A. triseriatus (Estimate: 0.1624; χ12=16.31, P < 0.0001) and TN (Estimate: 0.2627; χ12=4.44, P = 0.0351). TP was excluded from final models because it was always non-significant (P-values > 0.10).

Laboratory experiment

Estimated finite rate of population increase (λ′, P < 0.0001) and survival (P < 0.0001) both showed a significant interaction between species and tire leachate concentration. Aedes albopictus had similar λ′ than A. triseriatus in the control treatment (0 mg/L) but clearly divergent responses to increasing tire leachate concentration. Aedes albopioctus λ′ was consistently high and similar from 0–10,000 mg/L tire leachate but steeply declined to zero from 50,000 to 100,000 mg/L. In contrast, A. triseriatus λ′ declined at the lower concentration of 100 mg/L tire leachate, and was zero at 500, 10,000, and 100,000 mg/L (Fig. 1A). Survivorship results were broadly similar to λ′ for both species. Aedes albopictus survivorship declined steeply at 10,000 mg/L tire leachate whereas A. triseriatus survivorship declined steeply at 100 mg/L tire leachate (Fig. 1B). There was no A. albopictus survivorship at 100,000 mg/L tire leachate, whereas there was either very low or no A. triseriatus survivorship from 100 to 100,000 mg/L. Tire leachate concentration had no effect on development time (male: P = 0.6040; female = 0.3080) and mass (male: P = 0.9870; female = 0.2210) of both sexes, nor were any significant interactions of tire leachate with species (P-values > 0.1000) (Fig. 2). Aedes triseriatus were on average larger than A. albopictus for both sexes (male: P < 0.0100; female: P < 0.0001), and A. albopictus had faster female development than A. triseriatus (P < 0.0001) (Fig. 2). Development time for males did not vary between species (P = 0.3000).

Figure 1 Mean ± SE A. albopictus and A. triseriatusλ′ (A) and survivorship (B) exposed to varying concentrations of tire leachate in dose-response laboratory experiment.

The symbol for A. albopictus λ′ and survival at 100,000 mg/L tire leachate is hidden behind the corresponding symbol for A. triseriatus.

Figure 2 Mean ± SE A. albopictus and A. triseriatus female development time (days to eclosion) (A), male development time (days to elosion) (B), female mass (g) (C), and male adult body size (g) (D) exposed to varying concentrations of tire leachate in dose-response laboratory experiment.

Discussion

This study showed that tire leachate had negative effects on λ′ and survival of both the invasive mosquito A. albopictus and the native congener A. triseriatus, but that the invasive A. albopictus was clearly more tolerant. In field tires, A. triseriatus was more strongly negatively associated with zinc concentration compared to A. albopictus and another common co-occurring mosquito, C. pipiens. In our laboratory experiment, A. albopioctus λ′ was over 1.0, indicating positive population growth, and survival was high from 0–10,000 mg/L tire leachate, but both λ′ and survival steeply declined to zero from 10,000 to 100,000 mg/L. In contrast, A. triseriatus λ′ and survival declined at the lower concentration of 100 mg/L tire leachate, and was zero at 500, 10,000, and 100,000 mg/L.

The results of our study are inconsistent with the idea that tire leachate may facilitate the persistence of A. triseriatus after A. albopictus invasion by releasing compounds that are more toxic to A. albopictus. Instead, our data suggest that tire habitats may in fact support the spread of A. albopictus and the concomitant decline in A. triseriatus. Three of the five field sites that we sampled had tire water with mean zinc concentrations (1.35–2.39 mg/L) in excess of concentrations that caused a decline in A. triseriatus survival and λ′ (0 − 0.45 mg/L) in the laboratory experiment. This result, combined with the strong negative association between A. triseriatus and zinc concentrations in the field, suggests that A. triseriatus is likely to be negatively affected by tire leachate under many field conditions. Mean A. triseriatus λ′ was <1.0 across all tire leachate concentrations in our experiment, including in the control concentration that consisted only of DI water. This result suggests that our baseline experimental conditions were generally unfavorable to A. triseriatus, possibly because of food limitations. Nevertheless, the decline in A. triseriatus performance with even relatively small increases in leachate concentration was clear and not apparent in A. albopictus, which only showed declines in population performance in conditions higher than 8.9 mg zinc/L. Since its establishment in North America in the mid-1980s, A. albopictus has rapidly become the most common urban mosquito in many areas in the eastern United States. There are several explanations for this dominance; the most well documented being that A. albopictus is a superior competitor at the larval stage (Juliano, 2009; Smith, Freed & Leisnham, 2015) and asymmetric reproductive competition (Lounibos, 2007). Our study suggests that tire habitats may also facilitate the dominance of A. albopictus through this species’ superior tolerance to compounds that leach from tire rubber.

No tires that we sampled in the field had zinc concentrations in excess of 8.9 mg/L suggesting that A. albopictus may be minimally affected by tire leachate in the field. In our field study, we visited three dump sites, where tires had been illegally discarded, and two sites at auto-repair shops, where used tires had been stacked. Interestingly, the three dump sites had tires with zinc concentrations at least three times higher than the sites at auto-repair shops. Tires leach compounds as their rubber deteriorates, and the rate of deterioration is the result of the time and intensity of exposure to the combined effects of temperature, UV radiation, and humidity. These environmental variables cause discoloration, cracking, and splitting of tire sidewall tire rubber and oxidation of the steal belt (Andrady et al., 1998; Andrady, Hamid & Torikai, 2003; Yu et al., 2015). It is possible that the discarded tires at the illegal dumping sites were exposed to the environment longer than tires at auto-repair shops, which would likely be eventually sold or recycled. Future research needs to examine the rate of tire degradation and resultant leaching of tire materials (and associated compounds) under different environmental conditions, especially UV exposure, and the effects of tire leachate across a larger suite of mosquito species so that we might be able to predict field sites with favorable or deleterious leachate conditions. Numerous studies have related larval habitat variables to larval mosquito communities in discarded vehicle tires (e.g., Beier et al., 1983; Kling, Juliano & Yee, 2007; Yee, Kneitel & Juliano, 2010). Our study suggest that tire leachate should be another variable to consider to better predict vector distributions and disease risks.

Variation in λ′ of both species as a result of tire leachate appears to be primarily driven by survival. Interestingly however, despite an approximately four-fold decline of mean A. albopictus survival from 1,000 to 10,000 mg/L tire leachate, A. albopictus λ′ remained unchanged (Fig. 1B). This result was probably due a small but important concomitant increase in female mass from 1,000 to 10,000 mg/L tire leachate concentration (Fig. 2). An important limitation of past studies that have tested the effects of tire leachate on aquatic organisms, and of many toxicological studies, is that inferences on toxicological impacts are limited to individual parameters of fitness, and usually that of survival. Toxicological studies that focus solely on the effects on survival, yield limited inference of population impacts over multiple generations. Some prior experiments using λ′ have generated different conclusions for both λ′ and of survival, reiterating the importance of including an analysis of λ′ for a more thorough examination of environmental impacts on populations (Livdahl & Sugihara, 1984; Juliano, 1998). Further, λ′ is a more biologically meaningful measure of population performance than considering individual fitness parameters, such as survival, development time, and body size, separately, as it accounts for nonlinear interactions among these parameters (Livdahl & Sugihara, 1984). Thus, although the increase in A. albopictus female mass from 1,000 to 10,000 mg/L tire leachate was non-significant, it may still have an important effect in preventing a decline in A. albopictus λ′. Addressing parameters of adult fitness, such as female body size, is also important when considering vector mosquitoes since the adult life-stage is of public health importance and adult traits that affect disease transmission can be affected by the conditions of larval habitat. For example, recent modeling research has shown that smaller adult female Aedes have decreased probability of dengue infection, primarily because of they have reduced nutritional reserves and shorter lifespans, and hence live out the extrinsic incubation period and become infectious (Juliano et al., 2016).

To our understanding this is one of the first studies to rigorously examine the effects of tire leachate on the population performances of mosquitoes. Our study did not, however, address the mechanism by which sub-lethal concentrations of tire leachate might negatively affect individual mosquito larvae and differentially affect A. albopictus and A. triseristus. One likely mechanism that tire leachate may affect mosquito larvae is by changing their microbial food resource. Numerous studies have demonstrated decreased overall bacterial abundances and altered bacterial communities under increasing tire leachate concentrations (e.g., (Leff, McNamara & Leff, 2007; Vukanti et al., 2009)). Aedes albopictus is more efficient at harvesting microbial food resources primarily because it spends more time feeding and more efficiently converts these resources into biomass (Kesavaraju, Damal & Juliano, 2008; Kesavaraju, Khan & Gaugler, 2011). Overall decreases in microbial food or changes in the composition of microbial communities as a result of tire leachate toxicity might therefore be expected to more severely negatively affect A. triseriatus. Another mechanism that tire leachate could affect mosquitoes is by being directly toxic to the larvae, and individuals from different species may vary in their tolerance. Insects detoxify natural (e.g., tannins) and synthetic (commercial insecticides) toxins via a number of biochemical mechanisms, and among the most well documented are P450 monoxygenases (P450s) (Scott, 1999). It possible that selection for increased upregulation of P450s in A. albopictus than A. triseriatus allows A. albopictus to better tolerate tire leachate. Suwanchaichinda & Brattsten (2002) demonstrated increased induction of P450s in A. albopictus larvae exposed to the common tire compound, benzothiazole and its derivatives (BZTs). The focus of Suwanchaichinda & Brattsten (2002) was to test if the exposure to tire compounds, particularly BZTs, made A. albopictus mosquito larvae more tolerant to commercial insecticides through P450 induction, and the study found evidence that it did (Suwanchaichinda & Brattsten, 2002).

The effects of tire leachate on mosquitoes are likely to be complex and act at both the immature (egg and larval) and adult life-stages. The main goal of this study was to compare effects of tire leachate on the population performances of two co-occurring Aedes mosquito species, A. albopictus and A. triseriatus, and it showed clear interspecific differences. Additional research needs to examine the suite of density dependent and trait-mediated impacts related to vector mosquitoes, including the effects on population performances of a larger number of tire-inhabiting species, on community processes, such as competition, predation, parasitism and mutualism, and on vector competence across a range of disease systems.

Supplemental Information

Data S1 Field data

Click here for additional data file.

Data S2 Lab data

Click here for additional data file.

We thank R Pozzatti, N Kirchoff, and D Bodner for assistance in data collection, Kathryn M. Conko (USGS) for preparing the tire leachate and help planning the laboratory experiment, and SA Juliano for statistical advice.

Additional Information and Declarations

Competing Interests

Author Contributions

Data Availability

The authors declare there are no competing interests.

Oswaldo C. Villena analyzed the data, wrote the paper, prepared figures and/or tables, reviewed drafts of the paper.

Ivana Terry and Kayoko Iwata conceived and designed the experiments, performed the experiments, analyzed the data, wrote the paper, prepared figures and/or tables.

Edward R. Landa conceived and designed the experiments, contributed reagents/materials/analysis tools, wrote the paper.

Shannon L. LaDeau contributed reagents/materials/analysis tools, wrote the paper, reviewed drafts of the paper.

Paul T. Leisnham conceived and designed the experiments, analyzed the data, contributed reagents/materials/analysis tools, wrote the paper, prepared figures and/or tables, reviewed drafts of the paper.

The following information was supplied regarding data availability:

The raw data have been uploaded as Supplemental Files.

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
