# Peer review of "Effects of tire leachate on the invasive mosquito Aedes albopictus and the native congener Aedes triseriatus"

_PeerJ, doi:10.7717/peerj.3756_

## Round 0.1 · original submission · Minor Revisions

This is a very nice article that sheds light on the ecology of this invasive mosquito species and needs very small revisions (please, see the few suggestions made by the reviewers). Congratulations.

Reviewer 1 ·

Basic reporting

This manuscript describes a series of field and laboratory experimental manipulations of mosquitoes, to determine the effect of tire leachate and relative influence of a number of factors on egg and larval development. The experimental design and resultant data are informative and effectively address the stated aims of the work. Such studies are important for our understanding of not only the ecology of potential disease vectors, but the risk they pose in terms of disease transmission.Specific comments follow below:

1. In the Materials and Methods section, please provide the temperature and humidity ranges or environmental conditions of study sites.

2. Line 303; A. albopictus should be Aedes albopictus.

3. In the Discussion section, it would be good to have a statement that directly answers the question posed in the title.

4. Please check your references format carefully.

Experimental design

Very sound and clear. Just suggesting to add the study sites map could be good.

Validity of the findings

Interesting and important findings.

Additional comments

This manuscript describes a series of field and laboratory experimental manipulations of mosquitoes, to determine the effect of tire leachate and relative influence of a number of factors on egg and larval development. The experimental design and resultant data are informative and effectively address the stated aims of the work. Such studies are important for our understanding of not only the ecology of potential disease vectors, but the risk they pose in terms of disease transmission.Specific comments follow below:

1. In the Materials and Methods section, please provide the temperature and humidity ranges or environmental conditions of study sites.

2. Line 303; A. albopictus should be Aedes albopictus.

3. In the Discussion section, it would be good to have a statement that directly answers the question posed in the title.

4. Please check your references format carefully.

·

Basic reporting

I enjoyed reading this manuscript. It is written clearly, justified well, and is concise and unambiguous. Literature citations appear sufficient, and the content of the manuscript is professionally presented.

Experimental design

The design and technical components of this study are well designed, clearly articulated and well justified. Careful consideration has been given to the laboratory experiments used to complement the field findings. This is a valuable addition to the research and supports robust study conclusions.

Validity of the findings

I have no issue with the validity of study findings. The authors have identified a gap in the literature on mosquito development in used tires. The data and statistics are robust and the conclusions well supported by the results, without overstating their significance. The authors have been thoughtful in the implications and future directions of their study.

Additional comments

I have only minor suggestions to help the authors improve their study:
1. It was somewhat jarring in the abstract to switch from mg/L Zn to mg/L tire leachate. My recommendation is to make mg/L Zn the main unit in the abstract and only present mg/L tire leachate in parentheses.
2. It seems relevant that the authors should provide some justification in the ms for why they chose to omit Cx. pipiens (an important vector of WNV) from the same trials they subjected the other two mosquitoes.
3. L244-246. Can you provide an approximate date when A. albopictus became established in the US?
4. Please attend to grammatical errors on L296, L304 and L320.

---

## Round 0.2 · accepted · Accept

I think that the manuscript is now ready to be published.